# Nutritional Factors Associated with Late-Onset Sepsis in Very Low Birth Weight Newborns

**DOI:** 10.3390/nu14010196

**Published:** 2021-12-31

**Authors:** Juliany Caroline Silva de Sousa, Ana Verônica Dantas de Carvalho, Lorena de Carvalho Monte de Prada, Arthur Pedro Marinho, Kerolaynne Fonseca de Lima, Suianny Karla de Oliveira Macedo, Camila Dayze Pereira Santos, Saionara Maria Aires da Câmara, Anna Christina do Nascimento Granjeiro Barreto, Silvana Alves Pereira

**Affiliations:** 1Neonatal Intensive Care Unit, Maternidade Escola Januário Cicco, Universidade Federal do Rio Grande do Norte, Natal 59012-300, Brazil; julianycaroline@hotmail.com (J.C.S.d.S.); carvalho.anaveronica@gmail.com (A.V.D.d.C.); lorenamonte88@gmail.com (L.d.C.M.d.P.); arthurpmarinho@gmail.com (A.P.M.); kerolaynne.fonseca@gmail.com (K.F.d.L.); suianny_k@hotmail.com (S.K.d.O.M.); camila_nutufrn@hotmail.com (C.D.P.S.); annachristina.natal@hotmail.com (A.C.d.N.G.B.); 2Department of Physiotherapy, Universidade Federal do Rio Grande do Norte, Natal 59075-000, Brazil; saionara.aires@ufrn.br

**Keywords:** neonatal sepsis, enteral nutrition, parenteral nutrition, preterm infant, neonatal intensive care units

## Abstract

Background: Delayed onset of minimal enteral nutrition compromises the immune response of preterm infants, increasing the risk of colonization and clinical complications (e.g., late-onset sepsis). This study aimed to analyze associations between late-onset sepsis in very low birth weight infants (<1500 g) and days of parenteral nutrition, days to reach full enteral nutrition, and maternal and nutritional factors. Methods: A cross-sectional study was carried out with very low birth weight infants admitted to a neonatal intensive care unit (NICU) of a reference maternity hospital of high-risk deliveries. Data regarding days of parenteral nutrition, days to reach full enteral nutrition, fasting days, extrauterine growth restriction, and NICU length of stay were extracted from online medical records. Late-onset sepsis was diagnosed (clinical or laboratory) after 48 h of life. Chi-squared, Mann–Whitney tests, and binary logistic regression were applied. Results: A total of 97 preterm infants were included. Of those, 75 presented late-onset sepsis with clinical (*n* = 40) or laboratory (*n* = 35) diagnosis. Maternal urinary tract infection, prolonged parenteral nutrition (>14 days), and extrauterine growth restriction presented 4.24-fold, 4.86-fold, and 4.90-fold higher chance of late-onset sepsis, respectively. Conclusion: Very low birth weight infants with late-onset sepsis had prolonged parenteral nutrition and took longer to reach full enteral nutrition. They also presented a higher prevalence of extrauterine growth restriction than infants without late-onset sepsis.

## 1. Introduction

Late-onset sepsis remains a significant cause of morbidity and mortality in neonatal intensive care units (NICU). Its incidence ranges between 10% and 21% in very low birth weight (<1500 g) and between 34% and 41% in extremely low birth weight infants (<1000 g) [1]. Late-onset sepsis can be diagnosed after 48 h of life [2], mainly due to coagulase-negative staphylococci microorganisms [3].

The immature gastrointestinal tract and immune system of preterm infants are predisposed to infectious morbidity [4]. Prematurity is critical; thus, minimal enteral nutrition is the main contributor to microbiological programming [5] and immunological, metabolic, and gastrointestinal development, allowing transition to full enteral nutrition [6]. Delayed onset of minimal enteral nutrition compromises the immune response of preterm infants [7], decreasing the repair of intestinal epithelium and mucus production [8]. The gastrointestinal tract of preterm infants has a large and fragile surface area, filled by a thin epithelial cell monolayer that covers a highly immunoreactive submucosa [9]. This increases the risk of colonization because intestinal defense mechanisms and epithelial cells are immature and present exaggerated inflammatory responses to commensal and pathogenic bacteria [9,10].

Maturational weakness exposes newborns to clinical and functional complications, especially those with prolonged parenteral nutrition or admitted to NICU [11,12]. However, many studies highlighted the general risk factors for antenatal and postnatal sepsis associated with prematurity and nutritional variables, but not restricted to newborn weight [3,6,7,8,10]. In this context, we aimed to analyze associations between late-onset sepsis in very low birth weight infants and days of parenteral nutrition, days to reach full enteral nutrition, and maternal and nutritional factors.

## 2. Materials and Methods

This cross-sectional study was approved by the research ethics committee (CAAE 76397417.5.0000.5292) and conducted according to the Declaration of Helsinki. We included very low birth weight infants admitted to a NICU of a reference maternity school for high-risk deliveries, where 30% of all monthly deliveries (approximately 300) were preterm. Parents or guardians signed an informed consent form.

### 2.1. Sample Characterization

Data regarding very low birth weight infants admitted to the NICU between October 2017 and September 2018 were extracted from online medical records. We excluded infants with severe congenital malformations, inborn errors of metabolism, chromosome disorders, birth weight < 500 g or gestational age < 23 weeks, transferred from another hospital, deceased, or twins (more than two fetuses interfere with outcomes and worsen nutritional status at birth).

### 2.2. Variables

Late-onset sepsis was diagnosed (clinical or laboratory) after 48 h of life, according to the Brazilian Health Regulatory Agency [2]. NICU length of stay (in days) was registered between admission and discharge. Maternal variables, days of parenteral nutrition, days to reach full enteral nutrition, fasting days, progression to necrotizing enterocolitis (yes or no), extrauterine growth restriction, and days in NICU were extracted from medical records and included in data analysis.

Necrotizing enterocolitis was classified according to the modified Bell staging: IA (mild intestinal signs, such as gastric distension and increased pre-gavage residues), IIA (paralytic ileus and intestinal dilation), and IIIB (pneumoperitoneum) [13].

### 2.3. Protocol to Parenteral and Enteral Nutrition

Early (started in the first days of life) and optimized parenteral nutrition (2 to 3.5 g·kg^−1^·d^−1^ of protein) was administered with lipids (1.5 to 3 g·kg^−1^·d^−1^) at a glucose infusion rate of 4 to 6 mg·kg^−1^·min^−1^ (progression up to 10 to 12 mg·kg^−1^·min^−1^). Parenteral nutrition was prepared in the hospital pharmacy clean-room; transported in a 3-in-1 bag with emulsified proteins, carbohydrates, and lipids; and installed at night by the nursing staff (sterile procedure), preferably using peripherally inserted central catheters.

Minimal enteral nutrition was preferably initiated with the mother’s own breast milk by orogastric tube within the first 24 h with a volume of 10 to 20 mL·kg^−1^·d^−1^ every 3 h. When not available, pasteurized milk from a human milk bank or infant formula was used. Progression was performed at rate of 20 to 35 mL·kg^−1^·d^−1^ until reaching full enteral nutrition (120 mL·kg^−1^·d^−1^) and after hemodynamic stabilization, absence of signs of food intolerance, and use of breast milk additives (i.e., enteral nutrition with volume of ≥100 mL·kg^−1^·d^−1^).

We aimed to achieve full enteral nutrition in two weeks for preterm infants < 1000 g and in 7 to 10 days for preterm infants > 1000 g. Daily increases were performed in the first (10 to 20 mL·kg^−1^·d^−1^) and from the second week (up to 35 mL·kg^−1^·d^−1^). Full enteral nutrition was considered when infants reached 120 mL·kg^−1^·d^−1^ [14,15].

Days of parenteral nutrition were considered as the number of days between the beginning of intravenous nutrition and nutritional suspension due to intercurrence or complications (e.g., septic shock with metabolic acidosis and sepsis with paralytic ileus) or when patients achieved full enteral nutrition [14,15]. Fasting represented the number of days without enteral nutrition. Variables were categorized into parenteral nutrition (yes or no) and days to reach full enteral nutrition (≤14 and >14 days) [16,17]. Fasting days were categorized as ≤4 and >4 days, according to the day of energy reserve expected for infants since adverse health outcomes may occur after four days [18,19].

Extrauterine growth restriction was classified using the intergrowth curve at hospital discharge if infants reached a Z-score of weight ≤ −2 [17,20].

### 2.4. Data Analysis

Data normality was assessed using the Kolmogorov–Smirnov test. Pearson’s Chi-square test evaluated the association between late-onset sepsis, nutritional variables (parenteral nutrition, days to reach full enteral nutrition (≤14 or >14 days) [16], fasting days (≤4 or >4 days) [18,19]), and maternal variables (prenatal care, chorioamnionitis, urinary tract infection, and time of ruptured membranes) [2]. The Mann–Whitney test compared late-onset sepsis and continuous variables (birth weight, gestational age at birth, extrauterine growth restriction, days of parenteral nutrition, and days to reach full enteral nutrition), and median and interquartile range were presented according to the occurrence of late-onset sepsis. Multivariate binary logistic regression models evaluated variables associated with late-onset sepsis. All variables with *p* < 0.20 in the bivariate analysis were included in the regression model. The final model included only variables with *p* < 0.05. Statistical analysis was conducted using the Statistical Package for the Social Sciences (IBM^®^ SPSS^®^ software), New York, NY, USA.

## 3. Results

In total, 132 of 144 very low birth weight infants assessed in the study were included. Two infants were excluded due to malformation, four were quadruplets, and six were born with ≤500 g. Among those, 75 were males, and 75 had clinical or laboratory (*n* = 35) diagnosis of late-onset sepsis (6 had necrotizing enterocolitis, stage IIIb). Another 35 infants who died during hospitalization (4 with necrotizing enterocolitis) were excluded from the study, leading to a final sample of 97 participants. Infants with late-onset sepsis presented lower gestational age, lower Z-score at discharge, and more days to reach full enteral nutrition than infants with non-late-onset sepsis (Table 1).

Late-onset sepsis was associated with prolonged fasting, parenteral nutrition, and more days to reach full enteral nutrition. Regarding maternal variables, late-onset sepsis was associated with urinary tract infection. Table 2 and Table 3 present associations between late-onset sepsis and nutritional and maternal variables.

Maternal urinary tract infection and prolonged parenteral nutrition increased 4.86-fold and 4.24-fold the chances of late-onset sepsis, respectively (Table 4).

## 4. Discussion

This study demonstrated a high prevalence of late-onset sepsis in very low birth weight infants with prolonged parenteral nutrition and whose mothers had urinary tract infections. These preterm infants took longer to reach full enteral nutrition and presented higher prevalence of extrauterine growth restriction than those without late-onset sepsis. Parenteral nutrition is routine in NICU since reaching enteral nutritional needs is challenging [21].

The protocol used in this study guided early enteral progression and allowed monitoring by the pediatric gastroenterologist. However, the gastrointestinal system of preterm infants is still immature (i.e., reduced mucosal lining, gastric acidity, proteolytic enzymes, peristalsis, and increased permeability) and facilitates the transport of pathogens [22], hindering diet progression due to regurgitation, abdominal distention, or delayed gastric emptying [23]. Furthermore, this immaturity may favor bacterial translocation, reduce gastrointestinal hormone secretion, motility, and intestinal microvilli atrophy [24,25], increase the susceptibility of prolonged parenteral nutrition and fasting, and progress to enterocolitis [26,27,28,29].

In our sample, six preterm infants progressed to enterocolitis and reached stage IIIB on Bell staging; four died. Despite being a multifactorial condition, enterocolitis may occur between antenatal and post-birth [30] and progress to stage IIIB on Bell staging, imposing a severe condition that requires surgical intervention and diet suspension [31,32]. In a prospective multicenter study conducted with 926 preterm infants assigned to an initial diet, 51 (5.5%) developed necrotizing enterocolitis and 13 (26%) died [33].

Parenteral administration may also be considered a risk for late-onset sepsis [34]. In our protocol, the peripheral route was preferred for catheter insertion; however, the central catheter was indicated if (1) parenteral nutrition was expected for more than seven days or (2) nutritional needs of preterm infants exceeded the limits of peripheral administration (i.e., osmolarity > 900 and concentration > 12.5%) [34]. Catheter insertion may increase 2.56-fold the chance of acquiring late-onset sepsis [35] and duration of catheter use [36,37,38,39]. Corroborating our results, Sohn et al. (2001) reported a 5.7-fold increased risk of bloodstream infection in newborns who received parenteral nutrition [37]. Chances of nosocomial infections (4-fold) in infants under parenteral nutrition were also high in another study [38]. Despite identifying the potential risk of late-onset sepsis, previous studies were not restricted to newborn age, hindering comparison with our data [12,28].

In this study, preterm infants that evolved with sepsis took a median of 17 days to reach full enteral nutrition, longer than the infants without sepsis. Time is required for preterm infants to reach full enteral nutrition, increasing the chances of late-onset sepsis. According to Barreto et al. [40], episodes of late-onset sepsis are more prevalent when time to reach full enteral nutrition exceeds 10 days, NICU length of stay is prolonged, and Z-score at hospital discharge is low. Other studies also demonstrated that time to reach full enteral nutrition in very low birth weight infants was associated with sepsis [41,42]. In contrast, full enteral nutrition reached in 10 days reduced the risk of malnutrition 1.97-fold [43], and infants that initiated trophic feeding earlier reached full enteral nutrition faster than those who initiated late [44,45,46,47,48]. Another study [49] observed that full enteral nutrition was reached in 18.9 days, possibly due to late-onset trophic feeding (mean of 6.5 days) or the predominant type of food provided to preterm infants. Providing breast milk to preterm infants instead of formula was associated with short time to full enteral nutrition and low incidence of sepsis [48].

Regarding maternal variables, only urinary tract infection during pregnancy contributed to risk of sepsis in very low birth weight infants. Previous studies observed that urinary tract infection during pregnancy increased the chances of chorioamnionitis, premature labor, hospitalization, and death from sepsis [2,3].

Prematurity prolongs hospital length of stay in preterm infants < 32 weeks, regardless of clinical condition [48]. Additionally, clinical condition is more likely to worsen when accompanied by sepsis, increasing hospital length of stay (average of 44 days) [49]. An average of 79 days of hospitalization (95% confidence interval of 23 to 219 days) has also been reported [50]. Although our results were similar to previous studies [49,50] and hospital length of stay was inversely correlated with good nutritional evolution [51] and closely linked to extrauterine growth restriction [52], NICU length of stay was not included in the final regression model. This result might have been influenced by the number of preterm infants who died in our study. Therefore, clinical history, outcomes of preterm infants who died, and inclusion of a high-risk population (preterm infants < 1500 g) may be considered study limitations.

In this study, very low birth weight infants with late-onset sepsis presented higher prevalence of extrauterine growth restriction than those without late-onset sepsis. This result was probably due to the marked proteolysis of muscles and visceral proteins in preterm infants [53], which increased protein requirements and led to extrauterine growth restriction and complications related to a negative energy balance [54]. Other studies also showed that Z-scores from birth to discharge in preterm infants not small for gestational age were lower than observed in infants small for gestational age [54,55,56,57,58]. Furthermore, Barreto et al. (2012) found that more than half of low and extremely low birth weight infants (53.6%) were small for gestational age, and 89.3% had extrauterine growth restriction diagnosed at hospital discharge (median Z-score of −3.26), lower than observed in our study [40].

Although very low birth weight infants with late-onset sepsis fasted for four days, this variable was not included in the final regression model. Nevertheless, a previous study showed that fasting favors the proliferation of enteropathogenic microorganisms due to the absence of breast milk benefits, thus contributing to bacterial translocation [59]. Fasting may also lead to gastrointestinal atrophy if performed for two to three days [60]. We believe that early trophic feeding might have contributed to this result, decreasing the chances of proliferation of enteropathogenic microorganisms [44,45,46,47].

According to the literature, prematurity is also a well-known risk factor for late-onset sepsis [1,3,4,6]. However, this variable was not included in the final regression model probably because of the small variation in gestational age at birth between groups and the relatively small sample size.

This study is not free of limitations. Despite the robust sample, some infants lacked laboratory-confirmed sepsis. Although the literature rarely reports late-onset sepsis without laboratory confirmation, this criterion is maintained in low- and middle-income countries due to characteristics of health facilities. Therefore, this study is essential to control or reduce the incidence of sepsis in these countries. Lastly, clinical practices are essential for reaching early nutritional support.

## 5. Conclusions

Very low birth weight infants with late-onset sepsis had prolonged parenteral nutrition and took longer to reach full enteral nutrition. They also presented a higher prevalence of extrauterine growth restriction than infants without late-onset sepsis.

## Figures and Tables

**Table 1 nutrients-14-00196-t001:** Quantitative variables associated with late-onset sepsis (*n* = 97).

Variables	Late-Onset Sepsis	Non-Late-Onset Sepsis	*p*-Values
	Median (Q25;Q75)	Median (Q25;Q75)	
Birth weight (grams)	1110 (936.0;1300.0)	1282 (1175.0;1392.5)	0.001
Gestational age at birth (weeks)	29.5 (27.6;30.8)	30.1 (29.1;31.5)	0.012
Extrauterine growth restriction	−2.13 (−2.7;−1.6)	−1.62 (−2.3;−1.1)	0.021
Days of parenteral nutrition	14.0 (11.0;17.5)	9 (5.0;13.0)	<0.001
Days to reach full enteral nutrition	17.0 (13.5;20.5)	12.0 (9.0;14.3)	<0.001
NICU length of stay	37.0 (28.5;56.0)	18.0 (14.0;23.0)	<0.001
Fasting days	4.0 (2.0;6.0)	1.0 (1.0;2.0)	<0.001

GA, gestational age. NICU, neonatal intensive care unit. Q25;Q75, quartile 25; quartile 75. *p*-value for Mann–Whitney test.

**Table 2 nutrients-14-00196-t002:** Nutritional variables associated with late-onset sepsis (*n* = 97).

	Late-Onset Sepsis	Non-Late-Onset Sepsis	*p*-Values
	*n* (%)	*n* (%)	
Days of parenteral nutrition			<0.001
≤14 days	27 (50.9%)	40 (90.9%)	
>14 days	26 (49.1%)	4 (9.1%)	
Days to reach full enteral nutrition ^a^			<0.001
≤14 days	16 (30.2%)	32 (76.2%)	
>14 days	37 (69.8%)	10 (23.8%)	
Fasting days ^b^			<0.001
≤4 days	23 (43.4%)	37 (86.0%)	
>4 days	30 (56.6%)	6 (14.0%)	

*p*-value for Chi-squared test. ^a^ Two missing values. ^b^ One missing value.

**Table 3 nutrients-14-00196-t003:** Association between maternal variables and infants with late-onset sepsis (*n* = 97).

	Late-Onset Sepsis	Non-Late-Onset Sepsis	*p*-Values
	*n* (%)	*n* (%)	
Prenatal care ^a^			0.725
Yes	41 (93.2%)	38 (95.0%)	
No	3 (6.8%)	2 (5.0%)	
Chorioamnionitis			0.451
Yes	1 (1.9%)	2 (4.5%)	
No	52 (98.1%)	42 (95.5%)	
Urinary tract infection			0.005
Yes	24 (45.3%)	8 (18.2%)	
No	29 (54.7%)	36 (81.8%)	
Time of ruptured membranes ^b^			0.326
At labor	41 (77.4%)	32 (74.4%)	
<18 h	4 (7.5%)	7 (16.3%)	
≥18 h	8 (15.1%)	4 (9.3%)	

*p*-value for Chi-squared test. ^a^ Thirteen missing values. ^b^ One missing value.

**Table 4 nutrients-14-00196-t004:** Variables included in the final binary logistic regression model for late-onset sepsis.

	Odds Ratio (95% CI)	*p*-Value
Days of parenteral nutrition (>14 days)	4.24 (1.02 to 17.69)	0.047
Days to reach full enteral nutrition (>14 days)	3.95 (1.26 to 12.30)	0.005
Maternal urinary tract infection (yes)	4.86 (1.45 to 16.26)	0.010
Extrauterine growth restriction (yes)	4.90 (1.62 to 14.79)	0.018

CI, confidence interval. NICU, neonatal intensive care unit.

## Data Availability

The data presented in this study are available on request from the corresponding author. The data are not publicly available due to privacy restrictions.

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
