# Peer review of "Nutritional Factors Associated with Late-Onset Sepsis in Very Low Birth Weight Newborns"

_nutrients, 2021, doi:10.3390/nu14010196_

Round 1

Reviewer 1 Report

Dear Authors, 

You have contributed with an interesting study on the important subject of factors associated with sepsis in neonates. 

I have one general comment and that is regarding the language. The English is fine, and I believe the original report (probably written in Portuguese) is of solid quality but it seems in some instances that the message is lost in translation. To give one example, row 43-46, is very hard to understand the message, row 88 "no PN was discontinued"? and also eg row 32 refers to LOS I understand. I will give some specific comments below:

  • Row 32 - I suggest to write "The incidence of late-onset sepsis in very low...."
  • The methods part lack a lot of information that is needed to have a thorough discussion: peripheral/PICCline/Central catheter ? Only amino acids given as PN or together with carbohydrates and lipids? Three chamber bags or separate bottles? Mixed in pharmacy clean room or on ward ? This is important to know when discussing sepsis. 
  • Better description of the PN/EN protocol day by day? How is weaning done? Compliance to protocol? 
  • Row 79: 1500 ml/kg/d is an absurd amount,  pls correct. Only amino acid? What concentration and calculated in g AA/kg/day? please clarify
  • Discuss why some patients take longer time to reach full EN
  • Row 112 says 75 had clinical LOS and 34 of them was based on laboratory diagnosis. In the abstract it is also mentioned that 37 was based on clinical decision. 34+37=71. The other 4 patients?
  • The discussion (almost whole page 5) is only a literature review with little connection to the own findings. Especially weaning protocol, IV line and way of administration of PN (aseptic protocols etc) in the unit should be discussed in this context. 

Author Response

I have one general comment and that is regarding the language. The English is fine, and I believe the original report (probably written in Portuguese) is of solid quality but it seems in some instances that the message is lost in translation. To give one example, row 43-46, is very hard to understand the message, row 88 "no PN was discontinued"? and also eg row 32 refers to LOS I understand. I will give some specific comments below: Row 32 - I suggest to write "The incidence of late-onset sepsis in very low...."

Response: Thank you. We adjusted the entire sentence (lines 32, 43 to 46, and 88) and sent the manuscript for English revision.

  • The methods part lack a lot of information that is needed to have a thorough discussion: peripheral/PICCline/Central catheter ?

Response: We included this information in the “sample characterization”

  • Only amino acids given as PN or together with carbohydrates and lipids? Three chamber bags or separate bottles? Mixed in pharmacy clean room or on ward ? This is important to know when discussing sepsis. 

Response: The mix of parenteral nutrition was prepared in the pharmacy clean room and transported in a 3-in-1 bag with emulsified proteins, carbohydrates, and lipids. We included a more detailed description of the protocol in the methods section. If needed, we can also include the entire protocol as supplementary material.

  • Better description of the PN/EN protocol day by day? How is weaning done? Compliance to protocol? 

Response: We appreciate the suggestions. We included a more detailed description of the protocol in the methods section. We can also include the entire protocol as supplementary material, if needed.

Row 79:1500 ml/kg/d is an absurd amount,  pls correct. Only amino acid? What concentration and calculated in g AA/kg/day? please clarify .

Response: Thank you for the observation. We included a more detailed description of the protocol in the methods section.

  • Discuss why some patients take longer time to reach full EM:

Response: Thank you for the suggestion. We added new citations to justify this result in the discussion.

  • Row 112 says 75 had clinical LOS and 34 of them was based on laboratory diagnosis. In the abstract it is also mentioned that 37 was based on clinical decision. 34+37=71. The other 4 patients?

Response: Thank you for the observation. We corrected this information in the abstract and results section.

  • The discussion (almost whole page 5) is only a literature review with little connection to the own findings. Especially weaning protocol, IV line and way of administration of PN (aseptic protocols etc) in the unit should be discussed in this context. 

Response: Thank you for the suggestion. The entire discussion was remodeled and new hypotheses and citations were included to justify our results throughout the topic.

Reviewer 2 Report

Authors have analyzed the nutritional factors associated with late onset sepsis. There are various confounding variables which needs to be addressed before we derive any conclusion. Prematurity is a major one and antenatal factors were not mentioned. 

Methods: 

Line 78: what do the authors mean by "first hours of life"

Line 79: Maximum or minimum of ...... ?

Please rephrase Line 78 to 80. It is unclear whether authors are talking about AA or TPN or fluid goals? 

What day of life in which enteral feeds are started. For example a 25 weeker may not be ready for enteral feeds in the same way as 32 weeker. Your hospital/study protocol should be listed here. Similarly on the study cohort as continuous variable rather than < or > 4 days. Why was 4 days chosen as dichotomous variable. As you are providing parenteral nutrition how will energy reserve come into play in this?

Results:

Please provide the 25th and 75th centile values in brackets. 

NICU length of stay in days and length to reach full feeds - are these considered as continuous variables or dichotomous variables in the regression analysis. Please provide the full regression model. 

Table 2. Numbers don't add up for each variables? Kindly explain. 

How did you account for mortality? Were they censored from analysis?

How many had culture proven sepsis? Can you do sensitivity analysis based on this?

Did the authors adjust for prematurity as this is a significant confounder. 

How many babies had NEC/Medical NEC? How many had spontaneous perforation? Surgeries? 

Discussion : needs to be reshaped based on above findings. 

Author Response

  • Authors have analyzed the nutritional factors associated with late onset sepsis. There are various confounding variables which needs to be addressed before we derive any conclusion. Prematurity is a major one and antenatal factors were not mentioned.

Response: Thank you for the observation. The variable “gestational age” was significant in the bivariate analysis, but it was not included in the final model. Nevertheless, we recognized the need to include maternal variables and presented new tables with these results (Tables 3 and 4).

Methods:

  • Line 78: what do the authors mean by "first hours of life";

Line 79: Maximum or minimum of ...... ?

Please rephrase Line 78 to 80. It is unclear whether authors are talking about AA or TPN or fluid goals?

Response: Thank you for the observation. We included a more detailed description of the protocol in the methods section. We can also include the entire protocol as supplementary material, if needed.

  • What day of life in which enteral feeds are started. For example a 25 weeker may not be ready for enteral feeds in the same way as 32 weeker. Your hospital/study protocol should be listed here. Similarly on the study cohort as continuous variable rather than < or > 4 days. Why was 4 days chosen as dichotomous variable. As you are providing parenteral nutrition how will energy reserve come into play in this? )

Response: Thank you for the observation. We inserted a more detailed description of the protocol in the manuscript; however, we can also include the entire protocol as supplementary material, if needed.

Regarding categorization of the fasting period as ≤ 4 and > 4 days, we used the recommendation of studies Leite,H.P., and Unger,A. according to the day of energy reserve expected for infants since adverse health outcomes may occur after four days.

Results:

  • Please provide the 25th and 75th centile values in brackets.

Response: Thank you. Table 1 was adjusted.

  • NICU length of stay in days and length to reach full feeds - are these considered as continuous variables or dichotomous variables in the regression analysis. Please provide the full regression model.

Response: They were presented as quantitative variable. However, after your suggestions, a new model was built and this variable was not included in the final regression.

  • Table 2. Numbers don't add up for each variables? Kindly explain.

Response: Thank you for the observation. Deaths were excluded for this analysis. We adjusted table legends to clarify this point.

  • How did you account for mortality? Were they censored from analysis?

Response: The 35 infants who died during hospitalization were not included in the analysis showed in Table 2 (nutritional variables associated with late-onset sepsis). We added this information in Table 2, results, and discussion (limitation) sections.

  • How many had culture proven sepsis? Can you do sensitivity analysis based on this?

Response: Thirty-five preterm infants had culture-proven sepsis and 40 clinical diagnosis, according to the Brazilian Health Regulatory Agency. We included this information in the discussion section.

  • Did the authors adjust for prematurity as this is a significant confounder.

Response: Yes, we included the variable “gestational age” in the regression, but it was not significant in the final model.

  • How many babies had NEC/Medical NEC? How many had spontaneous perforation? Surgeries?

Response: We included this description in the results and discussion sections

  • Discussion: needs to be reshaped based on above findings.

Response: Thank you for the suggestion. We included new hypotheses and citations to justify our results throughout the discussion section.

Round 2

Reviewer 1 Report

Thanks for providing a much improved manuscript. All my previous issues have been properly addressed. 

Author Response

Thank you. We appreciate your comment.

Reviewer 2 Report

Authors have incorporated various changes suggested. However, still there are scopes for improvement. 

In abstract; Background: authors have to state the hypothesis rather than mentioning the association with various outcome variables.

Methods: please state all stages of NEC....because all will mandate keeping the patient NPO. This needs to be adjusted. 

What is the reason for keeping the babies NPO in patients with late onset sepsis. Unless NEC concerns, which is almost in most babies given the prematurity this is a serious confounder. Please provide numbers in both groups and adjust as appropriate in regression models.

Results:

Each table (n) has different total number of patients. If authors exclude the patients who died then it should be excluded from all tables. Table 2 it has been excluded but table 3 included and n is not mentioned in table 1. Please rectify this error. 

Cholestasis is multifactorial and not sure of its relevance. 

Please provide the entire regression model and not just positive findings. Prematurity is a well known to be associated with LOS. I don't understand how this is not significant. What do authors think about this. 

Discussion: Median is not same as average. Kindly change as needed.

Conclusion. Sentences "Prolonged NICU length of stay, low weight, and young gestational age at birth increased the prevalence of nosocomial infection and the risk for severe outcomes in preterm infants. Inadequate or delayed trophic feeding and late-onset sepsis are associated with growth deficit and neonatal development, increasing the risk of extrauterine growth restriction. " are not applicable. this study is not designed nor statistical tests were used to arrive at this specific conclusion. 

Round 3

Reviewer 2 Report

Authors have addressed the queries raised in prior revisions.